# Salient Concept-Aware Generative Data Augmentation

**Tianchen Zhao, Xuanbai Chen, Zhihua Li, Jun Fang, Dongsheng An,
Xiang Xu, Zhuowen Tu, Yifan Xing**

AWS DS3

## Abstract

Recent generative data augmentation methods conditioned on both image and text prompts struggle to balance between fidelity and diversity, as it is challenging to preserve essential image details while aligning with varied text prompts. This challenge arises because representations in the synthesis process often become entangled with non-essential input image attributes such as environmental contexts, creating conflicts with text prompts intended to modify these elements. To address this, we propose a personalized image generation framework that uses a salient concept-aware image embedding model to reduce the influence of irrelevant visual details during the synthesis process, thereby maintaining intuitive alignment between image and text inputs. By generating images that better preserve class-discriminative features with additional controlled variations, our framework effectively enhances the diversity of training datasets and thereby improves the robustness of downstream models. Our approach demonstrates superior performance across eight fine-grained vision datasets, outperforming state-of-the-art augmentation methods with averaged classification accuracy improvements by 0.73% and 6.5% under conventional and long-tail settings, respectively.

## 1 Introduction

Text-to-image (T2I) models [63, 68] have demonstrated remarkable success in generating high-fidelity images. One promising application is using synthetic image generation as a form of distillation, transferring knowledge into classifiers to improve their performance [69, 5, 7, 75, 97, 74]. However, T2I models often struggle to generate fine-grained categories due to the limitations of textual descriptions, introducing noise to the training set that can negatively impact the downstream classifier training. For example, ambiguous terms can lead to incorrect image synthesis, such as jaguar referring to either an animal or a car. Expert terminology outside the model's training vocabulary, such as class names from *iNaturalist* dataset, may produce unexpected output. Moreover, certain visual attributes, such as human identities, cannot be fully captured through text alone. Generative data augmentation (GDA) [99] addresses these limitations by conditioning synthesis on both text and image to better capture the subject characteristics.

Specifically, GDA leverages personalized image generation methods [96] to augment existing images of a given subject based on textual descriptions. However, existing approaches [76, 37, 83] rely on subject-specific optimization, which overly emphasizes instance-level details from the image prompts, limiting diversity when text prompts call for meaningful modifications. They also inherit issues from foundational tools such as DreamBooth [65] and Textual inversion [25], which are prone to overfitting or suffer from imprecise subject representations. On the other hand, approaches such as SuTI [14] provide more flexible alternatives, which are pre-trained on web-scale data and support zero-shot synthesis. However, these models often struggle to preserve the key concept in an image that defines its class, especially when the class definition is abstract or the object corresponding to the target concept on the image is small. This is caused by the inability of the methods to entail prior knowledge on what to focus on in the image. We are motivated to learn a specialized synthesis model tailored for

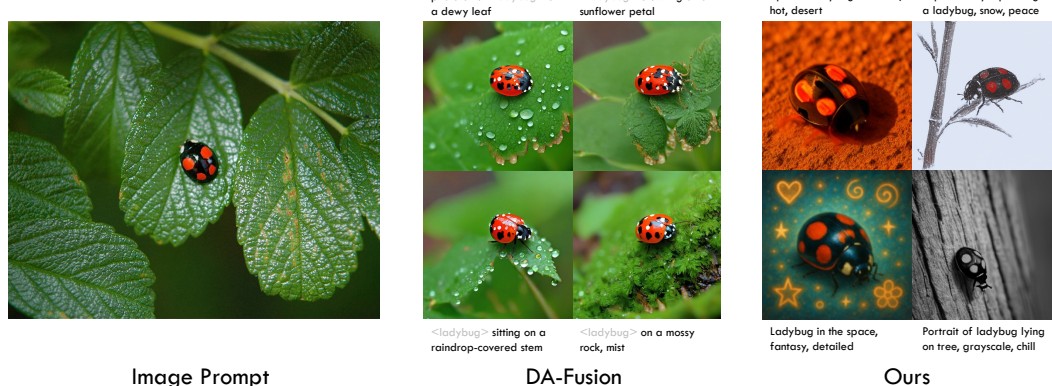

Figure 1: Comparison between our proposed data augmentation method with DA-Fusion [76]. DA-Fusion uses Textual Inversion [25] to learn the word embedding of the ladybug from the image prompt. But in this example, the generated outputs skew heavily toward a green color palette, failing to both preserve the ladybug's characteristics (black body with red spots) and align with the text prompts. Our method is designed with in-context understanding of the task: it selectively encodes salient concept (the ladybug) and effectively reduce the influence of irrelevant information (the greens and leaves) from the prompt, generating images with focused diversity and thereby effectively improves the robustness of the downstream models.

a target domain that understands domain-specific characteristics, intra-class variations, and inter-class relationships. This allows the model to generate novel, faithful, and diverse images that enrich the distribution of the original dataset, particularly for underrepresented or unseen categories, improving the robustness of the downstream classifier.

To this end, we define the problem of domain-specific personalized image generation for GDA: consider a joint distribution $(\mathbb{X}, \mathbb{Y})$, where each sample consists of an image $x_i \in \mathbb{X}$ and a corresponding fine-grained label $y_i \in \mathbb{Y}$, our goal is to generate image $x'_i \in \mathbb{X}$ that accurately captures the key concept about $y_i$ from the image prompt $x_i$, while also aligning with some relevant text prompt $t_i$ that describes $y_i$. We adopt the framework [14, 89] that uses an image embedding model to capture the concept of interest from image, which is then integrated with text embedding to control a diffusion model. While originally designed for personalized image generation [4, 86, 84, 26, 38, 29, 58], this approach has not been explored for GDA applications, particularly in properly balancing the fidelity-diversity trade-off.

Our insight is that the widely adopted practice of naively using image embeddings to control generative model is inadequate for GDA: these embeddings often entangle information that is less relevant to the concept of interest, restricting the model's ability to produce faithful results aligning with text prompts. To mitigate, we propose to train a salient concept-aware (SCA) image embedding model with the loss function that learns a discriminative representation space with high intra-class compactness, mitigating the entanglement of less relevant information, and inter-class separation, extracting distinctive salient concept characteristics from image prompts. As an example shown in Figure 1, embeddings learned from Textual Inversion capture irrelevant background information from the image, which conflict with the text prompt that asks for a different background. As a result, the generated images cannot accurately reflect the variants asked by the text prompt. In contrast, our generative model with salient concept-aware embedding focuses on the main subject and can generate images faithful to the subject. In addition, our approach is more convenient than methods that rely on additional input processing to capture target concepts, such as saliency detection or segmentation [42]. These approaches typically require additional human interventions to make sure the segmented concept is relevant to the task, and are limited to scenarios where the target object can be clearly defined with spatial boundaries [15, 88].

Our contribution is that we are the first to leverage the recent personalized image generation framework with dual embedding models for generative data augmentation, without the need for subject-specific optimization. The proposed framework mitigate the fidelity-diversity trade-off by training a salient concept-aware image embedding model that captures the salient concept of the image while adapting effectively to diverse text edit instructions. Our approach is evaluated on concept preservation, text alignment, and downstream classification accuracy across conventional, few-shot, long-tail, and out-of-distribution settings. Experiments on eight Fine-Grained Visual Categorization (FGVC)

datasets [17, 77, 56, 51, 79, 43, 41] demonstrate state-of-the-art performance over existing generative data augmentation methods. Specifically, our method improves classification accuracy by 0.73% in conventional settings and 6.5% in long-tail settings.

## 2 Related Work

Image generation has made remarkable progress with the success of foundation T2I diffusion models [55, 62, 63, 60, 20, 68, 6, 87, 19, 24, 90]. Moreover, their adaptability to incorporate additional controls has drawn significant interest [95, 98, 52, 54, 48, 40, 30, 36, 67, 71, 80], which are trained on paired image-text data at web scale but they struggle to establish concepts that are less frequently appeared in the training set or cannot be described by language clearly.

**Personalized Image Generation.** Different from image-editing approaches, personalized image generation faithfully reflects a specific concept in novel scenes, drawing inspiration from reference images from personal lives. Given a few samples of a concept, DreamBooth [65] fine-tunes the full model and Textual inversion [25] optimizes a word vector for the new concept, followed by a subsequent works generalized to muiltple concepts [44, 27, 4, 86]. To support broad adoption for practical usage, training-free methods [14, 89, 86, 46, 84, 70, 26, 38, 15, 13, 3] support zero-shot synthesis using reference images. These methods train a single model with large-scale database instead of per-object optimization, and do not necessarily preserve exact instance details from image prompt. However, users often observe suboptimal performance when working with these models, which are not specifically tailored to their categories of interest.

**Synthesis for Analysis.** Traditional methods such as Mixup [94] and CutMix [93] transform existing data but cannot introduce genuinely novel visual content. Recent efforts have focused on fine-tuning T2I models on in-domain datasets to improve fidelity [5, 73] or leveraging prompt enhancement techniques [92, 7, 45] to improve diversity. However, the scalability of synthetic data for classification remains unsatisfactory [28, 91, 21], as generated samples often exhibit limited transparency in their creation process, struggle to generalize across different contexts, and frequently require prompt engineering to obtain faithful results. Generative data augmentation (GDA) techniques [99, 76, 102, 53, 12, 82] leverage personalized image generation to improve the fidelity of synthetic data. DA-Fusion [76] utilizes SDEdit [52] and textual inversion to modify real samples with controlled generation strength. DiffuseMix [37] combine cut-mix with pix2pix [10] generated images. Diff-Mix [83] combines T2I personalization with I2I translation. Existing methods mostly use a combination of popular tools to edit upon real images by initializing diffusion with the images instead of noises, constraining image editing to a limited set of disentangled attributes such as textures, backgrounds, and colors. Our approach implements an end-to-end framework that encodes relevant image information into embedding, supporting concept-preserving image generation through free-form text editing. As a result, our method produces faithful, diverse training data that better challenges and improves the robustness of classifiers.

## 3 Our Approach

Generative data augmentation is a challenging task that aims to create diverse, high-fidelity datasets that improve the robustness and generalization capabilities of downstream classification models. Given a text-to-image generative model, a joint distribution $(\mathbb{X}, \mathbb{Y})$ representing the classification task, our goal is to fine-tune a domain-specific generative model $\mathcal{G}$ so that, given an image $x_i$ and a relevant text prompt $t_i$, the generated image $\mathcal{G}(x_i, t_i)$ satisfies: 1) concept consistency, where the generated image $\mathcal{G}(x_i, t_i)$ retains the key semantic attributes of $x_i$, and 2) text consistency, where $\mathcal{G}(x_i, t_i)$ accurately reflects the meaning of $t_i$.

Our main contribution lies in demonstrating the significance of the image embedding used for training synthesis model in achieving an optimal trade-off between text and image prompt consistency. Specifically, we divide our training into two stages. First, to achieve stronger supervision and better data efficiency, we independently train domain-specific salient concept-aware embedding model (SCA) , as introduced in Section 3.1. Then, we train the synthesis model, taking frozen image and text embeddings as inputs and only tune the adapters, as described in Section 3.2. We then explain our data synthesis approach for generating downstream classification training data in Section 3.3.

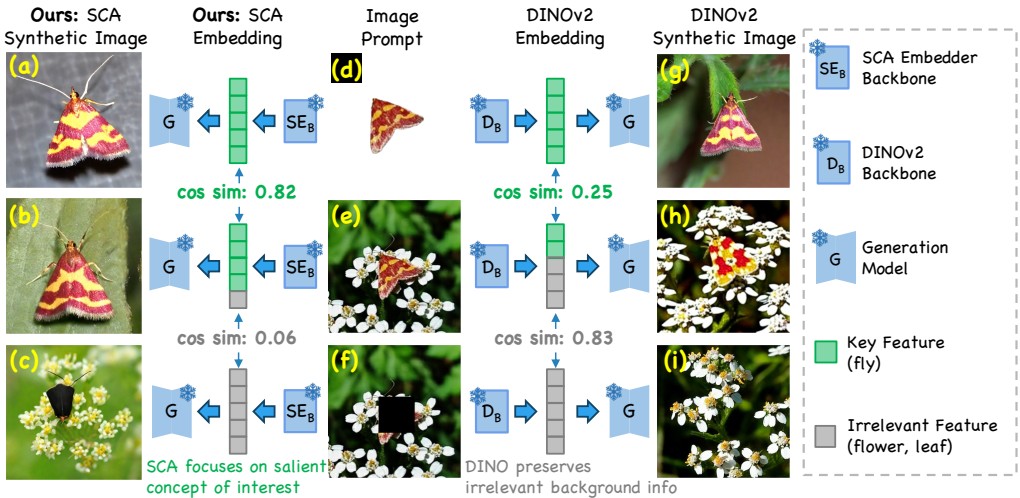

Figure 2: Comparison of generated images conditioned on various image prompts and null text prompt. Our model focuses on the fly in the image, evidenced by the high cosine similarity 0.82 between the segmented fly image (**d**) and the original fly image (**e**), allowing our synthesis model to generate high-fidelity fly images (**ab**). Foundation model like DINOv2 encodes lots of information about the background, evidenced by the high cosine similarity 0.83 between the embeddings of the original fly image (**e**) and the image with the fly removed (**f**), from which the model can reconstruct the background (**i**). Although DINOv2 works well on segmented images with the salient object to generate image (**g**), it introduces additional complexity in the pipeline, which is not suited for data augmentation purposes, and does not work well with abstract concepts without well-defined boundaries.

## 3.1 Salient Concept-Aware Embedding Model

Previous personalized image generation methods guide the synthesis with text and image feature extracted from large-scale pre-trained models such as CLIP [61] or DINO [11, 57]. However, as shown in Figure 2, those image features are entangled with irrelevant information for the salient concept from the prompt image, conflicting with the information provided by the text prompt [89].

Naive adoption of embeddings from a pre-trained classifier results in a significant performance degradation, as cross-entropy (CE) does not explicitly encourage discriminative capability under large intra-class or small inter-class appearance variations [18], and does not generalize well for underrepresented categories. As we demonstrate in our ablation study, data generated under the guidance of a classifier's embeddings provides marginal improvement for downstream training.

In this work, we introduce a penalty to improve both the intra-class compactness and inter-class discrepancy. Specifically, given samples $x_i$ and $x_k$ from class $p$, and $x_j$ from a different class $q$, where $p, q \in \mathbb{Y}$, we enforce the condition $D(f_i^p, f_k^p) < D(f_i^p, f_j^q)$, where $D$ is some distance measure in the embedding space and $f_i^p$, $f_k^p$, and $f_j^q$ are the respective embeddings, *i.e.*, $f = \mathcal{F}(x)$. To this end, we adopt the margin function:

$$\text{Term}_1 = e^{s \cdot \cos\left(\arccos(W_p^T f_i) + c\right)}, \text{Term}_2 = \sum_{q=1, q \neq p; j=1, j \neq i}^{|\mathbb{Y}|; N} e^{s \cdot W_q^T f_j},$$

$$\mathcal{L}_{\text{margin}} = -\frac{1}{N} \sum_{i=1}^{N} \log\left(\frac{\text{Term}_1}{\text{Term}_1 + \text{Term}_2}\right), \tag{1}$$

where $W_p \in \mathbb{R}^d$ denotes the $p$-th column of the weight matrix $W \in \mathbb{R}^{d \times |\mathbb{Y}|}$, $d$ is the embedding vector size, $|\mathbb{Y}|$ is the number of classes, $N$ is the total data volume, $W_p^T f_i$ represents the logit for the sample $x_i$, and variables $c$ and $s$ denote the margin penalty and scale factor, respectively. In practice, we normalize each individual weight $W_p$ and embedding feature $f_i$ in $\ell_2$. By adding an additive angular margin $c$ to the target angle, the embedding model $\mathcal{F}$ avoids encoding confounding characteristics from different classes and focusing more on extracting key concept of interest. As shown in Figure 2, the salient concept-aware embedding focuses on the main subject and can condition the generative model to output images faithful to the subject from image prompt.

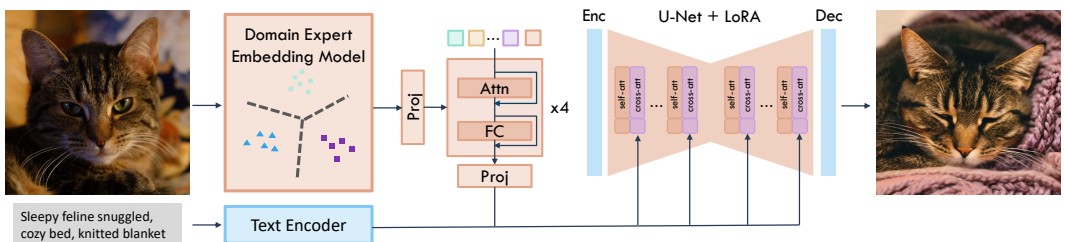

Figure 3: **Our proposed framework.** The SCA embedding model captures the salient concept and projects it with text embedding into cross-attention layers of U-Net. LoRA is applied to self-attention layers.

## 3.2 Synthesis Model

**Architecture.** Our salient concept-aware (SCA) embedding model trained in the first stage extracts visual features from image prompt, which are then projected into a feature space with the same dimensionality as the text features in the pretrained diffusion model. Similar to the text features $C^{\text{txt}}$, the projected image features $C^{\text{img}}$ are fed into every cross-attention layer in the U-Net, with new key and value projectors. The attention $Z$ is calculated as a weighted sum of the attention from both the text and the image embeddings, with a weight factor $\lambda$ set to the default value of 1:

$$Z = (QK^{\text{txt}})V^{\text{txt}} + \lambda(QK^{\text{img}})V^{\text{img}}. \tag{2}$$

Here, $K^{\text{img}}$, and $V^{\text{img}}$ are the key and value projections for the image prompt, defined as:

$$K^{\text{img}} = W_K^{\text{img}}C^{\text{img}}, \quad V^{\text{img}} = W_V^{\text{img}}C^{\text{img}}, \tag{3}$$

where $W_K^{\text{img}}$, and $W_V^{\text{img}}$ are weight matrices for key and value projections, respectively. For better adaptation with image embeddings, we apply LoRA [35] to the text branch:

$$Q = W_Q Z^{\text{prev}} + \Delta W_Q Z^{\text{prev}}, K^{\text{txt}} \quad = W_K^{\text{txt}}C^{\text{txt}} + \Delta W_K^{\text{txt}}C^{\text{txt}}, V^{\text{txt}} = W_V^{\text{txt}}C^{\text{txt}} + \Delta W_V^{\text{txt}}C^{\text{txt}}, \tag{4}$$

where $Z^{\text{prev}}$ is the previous state of the diffusion model, $C^{\text{txt}}$ denotes text prompt feature, $W_Q$, $W_K^{\text{txt}}$, and $W_V^{\text{txt}}$ are the weight matrices and $\Delta W_Q$, $\Delta W_K^{\text{txt}}$, and $\Delta W_V^{\text{txt}}$ are the corresponding LoRA linear layers, for the query, key, and value projections, respectively. We also apply LoRA to all self-attention layers of the U-Net in a similar fashion for better domain adaptation. To bridge between the embedding spaces of images and text, we adopt a light-weight image embedding projector [47] consists of four building blocks, each comprising a cross-attention layer and a fully connected layer with residual connections. See Figure 3 for more details on the synthesis model architecture.

**Multi-Modal CFG.** We apply classifier-free guidance by randomly dropping image or text prompt control during training. In particular for image prompt, instead of using zero vectors, we utilize the average embedding of training data as the class-conditioning input for unconditional models.

## 3.3 Synthesis for Analysis

**Data Construction.** For the training data used to fine-tune the adapter and LoRA, we generate corresponding prompts using image captioning models such as Blip2 [47] and MiniGPT [103]. For the inference data used to train the downstream classifier, we prompt Claude-3-Sonnet [2] as a separate large language model to produce contextualized descriptions of images featuring a specific class. These descriptions combine foreground objects and background elements in a concise manner. An illustrative example is provided below. Note that for both training and inference text prompts, we randomly replace the fine-grained class names with meta class names for robust training and diversified inference, *e.g.*, replace "bulldog" with "dog". To generate a synthetic data of a given category, we randomly sample real data from that category along with the corresponding LLM prompt to condition our synthesis model.

> ***Question****: Imagine there is a photo of [CLASS]. Describe the photo with one short sentence, followed by a few relevant descriptive terms about the background of the photo, separated by commas. This is used to prompt text-to-image models.*
> ***Answer****: [Baby] A cute baby lying on a soft blanket, nursery, pastel colors, toys.*
> *[Crane] A majestic crane stands tall, serene lake, lush greenery, distant mountains.*

| Backbone | Method | CUB | Aircrafts | Flowers | Cars | Dogs | Food | Avg. |
|---|---|---|---|---|---|---|---|---|
| RN50 | Vanilla | 86.62 | 89.14 | 99.29 | 94.47 | 87.15 | 92.01 | 91.45 |
| | CutMix [93] | 87.31 | 89.19 | 99.21 | 94.74 | 87.57 | 92.59 | 91.77 |
| | Mixup [94] | 86.74 | 89.43 | 99.49 | 94.41 | 87.49 | 91.45 | 91.50 |
| | Real-Guidance [28] | 87.26 | 89.21 | 99.26 | 94.64 | 87.27 | 92.03 | 91.61 |
| | Da-Fusion [76] | 86.47 | 87.96 | 99.31 | 94.69 | 87.36 | 91.59 | 91.23 |
| | Diff-Mix [83] | 87.50 | 90.12 | 99.44 | **95.21** | 87.88 | 92.21 | 92.06 |
| | Ours | **89.68** | **91.31** | **99.59** | 95.12 | **88.01** | **94.12** | **92.97** |
| ViT-B/16 | Vanilla | 89.94 | 85.46 | 99.53 | 94.23 | 91.26 | 94.11 | 92.09 |
| | CutMix [93] | 91.09 | 85.44 | 99.61 | 94.85 | 91.94 | **95.62** | 93.08 |
| | Mixup [94] | 90.89 | 86.27 | 99.70 | 94.87 | 91.83 | 94.17 | 93.12 |
| | Real-Guidance [28] | 90.11 | 85.13 | 99.56 | 94.67 | 91.86 | 93.59 | 92.49 |
| | Da-Fusion [76] | 89.97 | 83.84 | 99.58 | 94.55 | **91.88** | 94.23 | 92.01 |
| | Diff-Mix [83] | 91.13 | 88.07 | 99.74 | 94.93 | 91.76 | 93.53 | 93.19 |
| | Ours | **91.47** | **88.42** | **99.92** | 95.17 | 91.84 | 95.61 | **93.74** |

Table 1: Conventional classification Top-1 accuracy (%) across six datasets using RN50 and ViT-B/16. We achieve an average improvement of 0.73% over peer methods.

| Model | Method | IN | Aircrafts | Cars | Food | Flowers | Avg. |
|---|---|---|---|---|---|---|---|
| ViT-B/16 | Vanilla | 69.61 | 58.25 | 87.74 | 87.62 | 98.80 | 80.40 |
| | CutMix [93] | 70.93 | 61.27 | 89.05 | **88.93** | 98.97 | 81.83 |
| | Mixup [94] | 71.24 | 62.39 | 89.07 | 87.59 | 99.07 | 81.87 |
| | Da-Fusion [76] | 71.95 | 65.71 | 90.94 | 88.01 | **99.25** | 83.17 |
| | Diff-Mix [83] | 70.84 | 63.51 | 89.74 | 88.46 | 99.09 | 82.33 |
| | Ours | **72.41** | **72.44** | **92.72** | 88.38 | 99.17 | **85.02** |

Table 2: Few-shot classification Top-1 accuracy (%) using 10-shot training samples across five datasets.

**Filtering.** We use CLIP to assess text coherence by computing the cosine similarity between embeddings of text prompt and generated image. To evaluate salient concept coherence, we measure the cosine similarity between SCA embeddings of image prompt and generated image. Data samples with the lowest 10% in either text or salient concept consistency are filtered out.

**Classifier Training.** We pre-generate and filter synthetic training data, producing a final dataset that is of roughly $K$ times the size of the original training set. During classifier training, each batch combines synthetic and real samples, where $P\%$ of the data are sampled from synthetic pool.

# 4 Experiments

We introduce the experimental settings in Section 4.1. We compare our method with GDA baselines in Section 4.2. Ablation studies and qualitative analysis are shown in Section 4.3. More results are provided in the Supplementary.

## 4.1 Experimental Setting

**Generative Model.** For embedding model training, we fine-tune the model as defined in Equation 1. For synthesis model training, we only update image projector weights, image attention weights, and LoRA weights. We adopt classifier-free guidance (CFG) [32] with a strength of 5.0, and randomly apply conditional dropout to the embedding control and text control, with a 5% chance to drop the control from image, text or both, respectively. More details are provided in the Appendix.

**Synthesis for Analysis.** We generate synthetic images at 512 resolution and downsample them to 256 for storage, following diffusers [78] parameter settings. The synthetic dataset pool consists of approximately $K=5$ times the number of original training samples. For example, for a category with $n$ real samples, we generate $5n$ text prompts from LLM, and then produce the same amount of synthetic images of the same category. Under few-shot and long-tail settings, we ensure a more balanced class distribution by generating at least $250 (= 50 \times K)$ synthetic samples per class.

**Classifier.** We use CLIP pre-trained RN50 and ViT-B/16 with input resolution $224 \times 224$ as our backbone models, each topped with a 3-layer MLP projection head plus batch normalization. We apply RandAugment [16] while holding out Mixup and CutMix for comparison. In few-shot settings, we freeze the backbone and train only the projection head for 10% of the standard epoch count. Additional training parameters are detailed in the Appendix.

| Model | Method | IN-LT | Head | Mid | Tail | iNat | Head | Mid | Tail |
|---|---|---|---|---|---|---|---|---|---|
| | Vanilla | 69.49 | 83.12 | 64.96 | 39.82 | 65.52 | 76.54 | 68.31 | 59.25 |
| | CutMix [93] | 72.69 | 84.34 | 68.18 | 42.73 | 66.47 | 77.49 | 69.26 | 60.20 |
| | Mixup [94] | 70.73 | **84.38** | 66.22 | 41.28 | 67.59 | **77.61** | 70.38 | 61.32 |
| ViT-B/16 | Da-Fusion [76] | 71.96 | 83.61 | 70.45 | 44.63 | 66.61 | 77.51 | 69.35 | 60.27 |
| | Ours | **79.17** | 83.77 | **78.29** | **69.93** | **74.11** | 77.36 | **74.12** | **73.20** |
| | Vanilla + LWS | 72.28 | 79.78 | 69.83 | 56.77 | 71.97 | 74.23 | 73.45 | 69.83 |
| | Ours + LWS | **80.58** | **81.97** | **79.73** | **74.51** | **78.54** | **75.58** | **78.84** | **79.03** |

Table 3: Long-tail classification Top-1 accuracy (%). Results are reported separately for head, mid, and tail categories. Our approach outperforms existing data augmentation methods by 6.5%, especially for tail-classes. The performance of our method can be further improved through class re-balancing method such as LWS.

| Model | Method | L,L | W,W | L,W | W,L | Avg. |
|---|---|---|---|---|---|---|
| | Vanilla | 43.68 | 36.24 | 43.39 | 35.92 | 39.81 |
| | Real-Guidance [28] | 44.62 | 37.73 | 45.29 | 35.50 | 40.79 |
| ViT-B/16 | Da-Fusion [76] | 49.63 | 43.13 | 49.95 | 39.21 | 45.48 |
| | Diff-Mix [83] | 43.57 | 37.09 | 42.56 | 37.61 | 40.21 |
| | Ours | **54.86** | **45.53** | **56.51** | **47.74** | **51.16** |

Table 4: OOD classification Top-1 accuracy (%) on the Waterbird dataset. Our method is robust to unseen domain shifts by reducing spurious relations between background and foreground elements.

**Datasets.** We conduct experiments on widely used Fine-Grained Visual Classification (FGVC) datasets: *ImageNet*-1K [17] (IN), *iNaturalist*2018 [77] (iNat), Flower102 [56], Aircraft100 [51], CUB200-2011 [79], Cars102 [43], StanfordDogs120 [41] and Food101 [9]. Here, we clarify that we trained individual generation model for each dataset.

**Peer Methods.** We implement Mixup [94] and CutMix [93] with probabilities of 0.5 and 0.3 respectively. We also evaluate against GDA methods such as Real-Guidance [28], DA-Fusion [76], and Diff-Mix [83], following their official implementations. We also compare with long-tail recognition methods such as LWS [39] in the main text and others [100, 81, 34] in the Appendix.

## 4.2 Performance of Generative Data Augmentation

In this section, we evaluate the effectiveness of our approach across conventional, few-shot, long-tail, and out-of-distribution scenarios. The reported results are the averaged over three independent trials with different random seeds.

**Conventional Classification Performance.** In Table 1, we perform end-to-end training and evaluate the performance of classifiers trained with real and synthetic data compared to baseline methods on 6 FGVC datasets. Our method outperforms existing GDA techniques, achieving an average improvement of 0.73% on RN50 and 0.55% on ViT-B/16.

**Few-Shot Learning Performance.** We adopt an $N$-way 10-shot setting over three episodes, where $N$ is the number of classes. For each class, we randomly draw 10 samples from the training set, resulting in a total of $10N$ training data. We use this same limited dataset both for fine-tuning the generation model at 15% of the epochs compared to conventional setting and training the downstream classifier. Using fine-tuned generation model, we generate 250 synthetic images per category. During classifier training, we freeze the CLIP backbone and fine-tune classification head for 15 to 50 epochs, depending on $N$. As shown in Table 2, our method achieves the highest average performance across all datasets, achieving an improvement over existing methods by 1.85% on ViT-B/16.

**Long-Tail Classification Performance.** We follow the work of [49, 59] to conduct experiments on IN-LT and iNat. Our generation model is trained on the same dataset as the classification model. We generate at least 250 images for all categories for better class balancing. As shown in Table 3, our method achieves a remarkable performance improvement of 6.48% and 6.52% overall and 25.30% and 11.88% on tail categories for IN-LT and iNat, respectively. This improvement is more significant than in the few-shot setting. We hypothesize that this is because the synthesis model benefits from a larger number of training samples in the long-tail scenario, allowing it to better understand the relationships across categories and learning to generalize to capture salient concept for tail categories. As a result, downstream classifier learns more robust representations for the tail class by training on

| Dataset | Synthesis Model | DreamSim↓ | Vendi↑ | CLIPT↑ | Gen. Top5 |
|---------|-----------------|-----------|--------|--------|-----------|
| IN | Real-Guidance [28] | 0.4709 | **17.22** | **0.3396** | 88.27% |
| | Da-Fusion [76] | 0.4531 | 13.26 | 0.2801 | 76.93% |
| | Diff-Mix [83] | 0.4399 | 15.81 | N/A | 91.27% |
| | Ours | **0.3930** | 16.76 | 0.3119 | **98.84%** |
| iNat | Real-Guidance [28] | 0.5370 | 34.27 | **0.3302** | 31.09% |
| | Da-Fusion [76] | 0.5426 | 28.19 | 0.2387 | 27.60% |
| | Diff-Mix [83] | 0.5224 | 31.57 | N/A | 45.34% |
| | Ours | **0.4293** | **34.48** | 0.3106 | **87.48%** |

Table 5: Comparison of quality of the generated image from different DA methods. We use DreamSim to measure the foreground subject consistency and Vendi embedding score for measuring generated image diversity. We further use Clip to measure the consistency between input text prompt and the output image and evaluate the accuracy of external classifier on the generated images (Gen. Top5). Real-Guidance uses T2I generation models and therefore has the best text-image consistency. Our method achieves the overall best trade-off between salient concept consistency and diversity, particularly for fine-grained dataset such as iNat.

| Data | Model | DreamSim↓ | CLIPT↑ | Gen. Top5 | Cls. Top1 |
|------|-------|-----------|--------|-----------|-----------|
| iNat | Ours w/o SCA (CE) | 0.4626 | 0.2998 | 79.48% | 66.24% |
| | Ours w/o SDXL(+SD15) | 0.4587 | 0.2785 | 75.79% | 71.84% |
| | Ours w/o LoRA | 0.4506 | 0.286 | 80.71% | 68.86% |
| | Ours w/o Adapter(+MLP) | 0.4335 | 0.3048 | 87.17% | 73.43% |
| | Ours w/o Class Cond. | 0.4321 | 0.3092 | 86.44% | 73.77% |
| | Ours | **0.4293** | **0.3106** | **87.48%** | **74.01%** |

Table 6: Ablation study of the contribution of individual components in our synthesis pipeline on the iNat dataset. Here, Gen. Top5 measures the accuracy of external classifier on the generated images, and Cls. Top1 measures the accuracy of downstream classifier trained on the generated data. We selectively removed or replaced the proposed components to demonstrate their respective contributions. In particular, replacing the SCA embedding with classifier embedding significantly degrades performance.

more faithful and diversified data. We will discuss more about the generalization capability of the synthesis model in the next section.

We further apply learnable weight scaling (LWS) [39] that fine-tunes additional weight scaling factors for 40 epochs to improve the performance. At this stage, the model is trained on a balanced dataset consists of class-balanced sampling from the real dataset with probability $(1 − P)\%$ and instance-balanced sampling from the synthetic dataset with probability $P\%$. The performance can be further improved by 1.41% and 4.43% for IN and iNat, respectively. We also conduct experiments on other re-balancing and calibration methods in the long-tail recognition literature [100, 81, 34]. We find that LWS works the best for our settings and we will present the results for others in the Appendix.

**Performance under OOD Setting.** We conduct out-of-distribution (OOD) evaluation using the Waterbird dataset [66], which is created by superimposing foregrounds from the CUB dataset onto backgrounds from the Places dataset [101]. The results of our OOD evaluation are presented in Table 4, where "L" and "W" denote land and water, respectively. For example, "L,W" means a landbird placed against a water background. We train the model following $N$-way 5-shot scenario over three episodes using the CUB dataset, augmented with synthetic data generated through our GDA framework. Our approach is well-suited for this challenge, outperforming peer approaches by 5.68%, as our inference data construction in Section 3.3 explicitly incorporates diverse background variations, allowing classifier to mitigate spurious correlations with background elements.

### 4.3 Ablation Studies

**Performance of Personalized Image Generation.** In this section, we directly assess the diversity and faithfulness of our method's generated data and compare it with peer methods. For concept preservation, we use external expert classifiers from `timm` [85] leaderboard that achieve best recognition accuracy on target image domains. We report classification performance using Top-5 accuracy on synthetic images. To evaluate text-image alignment, we compute the cosine similarity between feature embeddings extracted from the CLIP-B/16 model's text and image encoders for input text prompts and generated images, respectively. We additionally use *DreamSim* [23] for measuring foreground subject consistency and *Vendi* [22] embedding score for measuring generated image diversity. The evaluation is conducted on image-text prompt pairs explained in Section 3.3 used for training

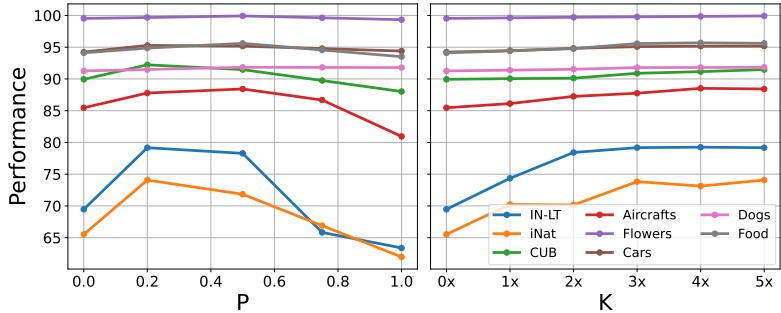

Figure 4: Impact of synthetic-to-real data ratio $K$ and synthetic sampling probability $P$ on Top-1 classification accuracy. Performance improves when a balanced proportion of synthetic data is used, but degrades when synthetic samples dominate. Increasing $P$ improves dataset diversity, but the benefits diminish as $P$ grows.

downstream classifiers, without applying any filtering. We present performance results in Table 5. The dataset generated by our method is diversified, achieving decent vendi and text consistency scores, and faithful, achieving best dreamsim scores and evaluation classifier accuracy. The advantage is significant for more fine-grained dataset like iNat, where we outperform the second-best model by 42.14% in Top-5 accuracy.

We intentionally exclude metrics such as CLIP image embedding consistency and widely used measures like FID [31] and KID [8] scores that favor identity mapping, which do not align with our objective of generating images that are meaningfully different from the input. For example, FID is minimized when comparing two identical datasets because it quantifies the distance between two distributions.

**Effectiveness of Components.** To assess the effectiveness of the SCA embedding, we train a classifier using a standard cross-entropy (CE) loss for image prompt feature extraction. As shown in Table 6, this results in a performance drop across all metrics, meaning that conventional CE-based embeddings are inadequate for preserving key concepts and generating diversified augmentations. Additionally, we examine the impact of key architectural choices in our generation framework. Specifically, we replace the SDXL backbone with Stable Diffusion 1.5 (SD15), substitute the adapter with a naive MLP, remove LoRA, and replace the image classifier-free null guidance with a zero vector. Across these ablations, we always observe performance drop across all metrics. This shows that each part of our architectural design is integral in improving the overall performance.

**Synthetic Ratios and Sampling Probabilities.** We conduct an ablation study on two key factors affecting downstream classification performance: the synthetic-to-real data ratio $K$ and the probability of sampling synthetic data $P$ during training, both defined in Section 3.3. In Figure 4, we analyze the Top-1 accuracy of the downstream classifier across different values of $K = \{0, 0.2, 0.5, 0.75, 1.0\}$ and $P = \{0, 1, 2, 3, 4, 5\}$. Our default settings use $K = 0.2$ for IN and iNat and $K = 0.5$ for other FGVC datasets, with $P = 5$. The results indicate that performance improves when $K$ is within the range of 0.2 to 0.5, but degrades when $K$ becomes excessively large. Incorporating synthetic data improves model generalization, but they may also introduce distributional shifts that negatively impact classification performance. Similarly, the sampling probability $P$ controls the diversity of the training set. Performance increases as $P$ grows but stabilizes beyond $P = 3$.

**Computational Cost Analysis.** A practical consideration for any data augmentation method is its computational overhead. Our approach involves two training stages: (1) training the SCA embedding model, and (2) fine-tuning the synthesis model using SCA embeddings. We provide a detailed analysis of the computational requirements for both stages. Training the SCA embedding model requires approximately 20% of the time needed to fine-tune the synthesis model. Table 7 reports wall-clock training times on an 8×A100 GPU setup across datasets of varying scales. For typical FGVC datasets with approximately 10K images, the total training time is 20-30 minutes. For larger-scale datasets such as iNat2018 (440K images) and ImageNet-1K (1.3M images), training requires 19 and 25 hours, respectively. For the largest dataset we experimented with, iNat2021 containing 2.7M images, training takes approximately 3 days. We argue that this computational cost is practical for most application scenarios, especially in domains where curating high-quality in-domain data is prohibitively expensive or time-consuming. Once trained, the synthesis model can generate unlimited augmented samples

| Dataset | FGVC | iNat2018 | ImageNet-1K | iNat2021 |
|---|---|---|---|---|
| Dataset Size | ~10K | 440K | 1.3M | 2.7M |
| Training Time | 20-30 mins | 19 hours | 25 hours | ~3 days |

Table 7: Training time for our method across datasets of different scales ($8\times$A100 GPUs). Training time scales reasonably with dataset size, ranging from 20-30 minutes for typical FGVC datasets to approximately 3 days for the largest dataset (iNat2021 with 2.7M images).

efficiently during downstream classifier training, providing consistent performance improvements over baseline GDA methods with a one-time training investment.

## 5 Conclusion

In this paper, we introduce a new approach to generative data augmentation (GDA) using a personalized image generation framework with a salient concept-aware (SCA) image embedding model, mitigating the fidelity-diversity tradeoff. SCA effectively captures key information from image prompts while preserving the semantics specified by text prompts, maintaining intuitive alignment across both modalities. Our method generates training data with focused diversity that significantly improve the robustness of the model, especially under long-tail settings. Experiments across eight fine-grained datasets show that our method outperforms existing GDA methods, particularly under long-tail and out-of-distribution settings. Through ablation studies, we compare the quality and diversity of our generated images with peer GDA methods, validate the effectiveness of each component in our synthesis pipeline, and studied the impact factors on the downstream classification performance. Our method is proved to be a promising approach for generative data augmentation.

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

# A    Model Preliminaries

**Synthesis Model Preliminary.** Diffusion models [33] can generate high-fidelity images from noise in an iterative manner, which is often accelerated with fast samplers like DDIM [72]. For conditional diffusion models, techniques like classifier-free guidance [32] are employed to balance image fidelity and sample diversity, realized by randomly dropping the condition during training. In this study, we adopt Stable Diffusion XL (SDXL) [60] as our base architecture. As a latent diffusion model [63], SDXL built upon UNet [64] with attention layers, conditioned on text features extracted from two frozen CLIP text encoders.

**Synthesis Model Size Statistics.** SDXL is our baseline text-to-image model with 3.856 billion parameters. SDXL-UNet and SDXL-VAE are the U-Net and VAE components with 2.955 billion and 84 million parameters, respectively. SDXL-TextEncoders are the two text encoders proposed in the original SDXL paper with a total of 818 million parameters. Adapter is our proposed module for fusing text and image representations, with 73 million parameters. LoRA is a technique for efficiently adapting new concept knowledge into pre-trained models, with a total of 387 million parameters. The Image Embedder is our module for embedding image representations, with 87 million parameters. Our adapter plus LoRA approach adds a total of 460 million parameters to the base SDXL model.

**Classifier.** For all experiments, we employed the official CLIP architecture with RN50 and ViT-B/16 backbones.

# B    More Training Details

**SCA Training.** Our SCA embedding model is developed from a pre-trained DINOv2-Base backbone [57], connecting with a head component with two linear layers containing batch normalization, and one fully connected layer. We fine-tune the whole SCA model with $s$ equals to 32, $c$ equals to 0.35. Following [1], we set the training epoch as 32, learning rate as $4e$-5, and batch size as 128, train with $8 \times$A100 GPUs using AdamW [50] optimizer. After training, we normalize the embedding from the backbone and feed it into our diffusion model.

**SDXL Training.** Diffusion model backbone is SDXL [60] implemented in `Diffusers` [78] repository, a model originally trained on high-resolution images of size $1024$. To optimize memory usage and accomodate lower resolution training data, we centercrop the image followed by resizing to a size of 512, allowing an effective batch size of 64 when distributed across $8 \times$ A100 GPUs. We use Blip2 [47] and MiniGPT-4 [103] to generate textual prompts for each training image. During the training process, to encourage diversity and improve the model's robustness in understanding prompts, we randomize the training prompts by splitting the generated prompt paragraphs into short sentences and key terms, which are then randomly sampled and recombined to form the final training prompts. Furthermore, we adopt classifier-free guidance [32], randomly applying conditional dropout to the embedding control and text control, with a 5% chance to drop the control from image, text or both, respectively. We also apply data augmentation techniques such as horizontal flipping and color jittering. We use the AdamW [50] optimizer with a constant learning rate of $1e$-5, where we update only the image projector weights, image attention weights, and LoRA weights. We train the adapted model for 10 epochs across all datasets. We observe that smaller datasets converge faster, while larger datasets converge slower. For example, it takes approximately 3 days to converge with $8 \times$A100 GPUs for iNaturalist [77] when training on a set of about 2.7 million images.

The training objective is to reconstruct the image prompt given the caption prompt. Taking ImageNet1K (IN1K) as an example, for training, the image prompts were sourced from the training split, and the corresponding text prompts were generated by the publicly available Blip and MiniGPT models. We forward pass the image $x$ through SCA and and text through language encoder to obtain respective embeddings $z_{image}^{SCA}$, $z_{text}$, which are taken by the synthesis model as inputs, and the image $x$ being the ground truth target for the output. Namely, training triplet is consisting of ($z_{image}^{SCA}$, $z_{text}$, $x$). Since the $\epsilon$ loss in the training phase do not require us to generate completed images, we only have the generated images available during the evaluation phase, where the image prompt and text prompt are the inputs, and generated image is the output.

**Classifier Training.** We primarily followed the training protocol established in Azizi et al. [5], with modifications tailored to our specific requirements. For ResNet50, we train *ImageNet*-1K and *iNaturalist* 2018 for 300 and 450 epochs respectively, and for smaller datasets we use 100 epochs;

| Model | Method | IN-LT | Head | Mid | Tail | iNat | Head | Mid | Tail |
|-------|--------|-------|------|-----|------|------|------|-----|------|
| ViT-B/16 | CE | 79.17 | **83.77** | 78.29 | 69.93 | 74.11 | **77.36** | 74.12 | 73.20 |
| | cRT [39] | 80.41 | 82.56 | 79.48 | 72.79 | 77.14 | 76.21 | 76.86 | 77.67 |
| | LWS [39] | **80.58** | 81.97 | **79.73** | **74.51** | **78.54** | 75.58 | **78.84** | **79.03** |
| | LAS [100] | 79.96 | 82.1 | 78.86 | 72.84 | 76.35 | 75.07 | 76.23 | 76.81 |
| | RIDE-3 [81] | 78.95 | 79.85 | 78.41 | 73.27 | 75.05 | 72.66 | 75.35 | 75.37 |
| | PC Softmax [34] | 79.98 | 81.52 | 79.04 | 73.96 | 76.62 | 74.85 | 76.09 | 77.63 |

Table 8: Comparisons among long-tail recognition methods, reported in classification Top-1 accuracy (%). The performance of the classifier can be further improved by fine-tuning the header with re-balancing or calibration approaches.

ViT-B/16 requires 60% more epochs across all datasets. The learning rate decays by a factor of 10 at 50% and 75% of total epochs. More precisely, for example, for large-scale datasets like *ImageNet*, we consider the following settings: for the ResNet50 model, training goes for 300 epochs with an input size of $224 \times 224$ and a large batch size of 8192. We use SGD optimizer with a learning rate of 3.6, a cosine decay schedule and a weight decay of $1 \times 10^{-4}$. There's a 5-epoch warmup phase. On the other hand, the ViT-B/16 model is trained for 480 epochs with the same input size but a smaller batch size of 1024, using the AdamW optimizer with a lower initial learning rate of $1 \times 10^{-3}$, also decaying via a cosine schedule but with a higher weight decay of 0.05. Its warmup period is longer at 10 epochs, and it additionally uses a drop path rate of 0.2. Both models apply a standard random cropping of $224 \times 224$ pixels, a horizontal flip probability of 0.5, label smoothing with a factor of 0.1, and RandAugment set to 1 for data augmentation. For FGVC datasets, we reduced the training duration to one-third of the epochs. We further adjusted the hyperparameters for few-shot learning and OOD detection tasks to optimize performance.

## C  Long-Tail Recognition Methods

Our method exhibits compelling performance under the long-tail setting. We further apply long-tail recognition methods upon our generated data to investigate potential performance improvements. Using the representation from the CE pre-trained model, we fine-tune for an additional 20-50 epochs during the second stage training on the classifier header (or experts for RIDE). For re-balancing approaches, we implement class balance strategies for real and synthetic data separately, as the latter is more balanced by construction. More precisely, the stage-two balanced dataset consists of class-balanced sampling from the real dataset with probability $(1 - P)\%$ and instance-balanced sampling from the synthetic dataset with probability $P\%$. For calibration approaches, we employ a heuristic approach that recalibrates the number of data points $n$ to be a weighted average of real and synthetic data: $n = \alpha n_{\text{real}} + (1 - \alpha) n_{\text{fake}}$, where $\alpha = 0.8$. The results presented in Table 8 demonstrate that our method achieves superior performance when combined with additional re-balancing or calibration methods. In particular, learnable weight scaling (LWS) achieves the best results compared to other methods for both datasets, outperforming the baseline by 1.41% and 4.43%.

## D  Broader Applicability Beyond Fine-Grained Classification

Although our main experiments focus on fine-grained visual classification, this choice is motivated by practical challenges in niche domains where expert models must be trained on domain-specific datasets that are often small or class-imbalanced. GDA is particularly valuable in these settings, as it can generate faithful and diverse samples by leveraging world priors learned from synthesis models, thereby improving robustness of downstream classifiers. We note that our current synthesis pipeline, built on SDXL, is optimized for object-centric image generation and may not be directly suitable for complex vision tasks such as text-rich VQA or GraphQA. Nevertheless, we demonstrate the broader applicability of our method through two additional experimental settings: abstract scene understanding and multi-modal visual classification.

### D.1  Abstract Scene Understanding: Places365

To evaluate performance beyond object-centric datasets, we conduct few-shot experiments on Places365 [101], a large-scale dataset for scene recognition. We follow the same N-way 10-shot

| Method | Vanilla | CutMix | Mixup | Da-Fusion | Diff-Mix | Ours |
|--------|---------|--------|-------|-----------|----------|------|
| Top-1 | 35.32 | 36.23 | 35.53 | 36.57 | 37.53 | **39.06** |
| Top-5 | 64.15 | 65.62 | 64.47 | 65.88 | 66.91 | **68.69** |

Table 9: Classification accuracy (%) using 10-shot training samples on Places365 validation set. Our method achieves consistent improvements over baseline augmentation methods.

| Method | CUB | Aircraft | Flowers | Cars | Dogs | Food | Average |
|--------|-----|----------|---------|------|------|------|---------|
| Zero-shot | 79.44 | 58.78 | 82.91 | 87.58 | 78.48 | 90.31 | 79.58 |
| Fine-tuned (Diff-Mix) | 89.37 | 66.92 | 94.71 | 95.30 | 87.14 | 83.23 | 86.11 |
| Fine-tuned (Ours) | **90.49** | **79.31** | **95.30** | **96.57** | **87.33** | **91.53** | **90.09** |

Table 10: Classification accuracy (%) under N-way 10-shot setting with Qwen2.5-VL-7B backbone. Our method demonstrates consistent improvements over both zero-shot and Diff-Mix augmented fine-tuning.

experimental protocol described in Table 2 of the main paper, with evaluation performed on the validation set. As shown in Table 9, our method outperforms the strongest baseline (Diff-Mix) by 1.53% and 1.78% in Top-1 and Top-5 accuracy, respectively. This demonstrates that our approach generalizes effectively to abstract scene understanding tasks beyond fine-grained object classification.

## D.2    Multi-Modal Visual Classification

Classification remains an important task for modern multi-modal models, which often require fine-tuning on domain-specific datasets to achieve production-level performance. To demonstrate the effectiveness of our generation method in this context, we explore multi-modal visual classification by performing supervised fine-tuning on a multi-modal large language model using FGVC datasets.

We use Qwen2.5-VL-7B as our backbone model. During fine-tuning, we tune only the vision projector and vision encoder while keeping the LLM frozen. The classification problem is reformulated as a 26-choice multiple-choice QA task, where the ground-truth label is always included among the choices. We use the following prompt template:

> *"You are an expert classifier. Based on the visual evidence in the image, select the most appropriate category. Please choose ONE of the following: A. beef_tartare B. creme_brulee ... Z. tacos. Please respond ONLY with your answer wrapped in this format: <answer> LETTER </answer>."*

Table 10 presents classification accuracy under the N-way 10-shot setting across six FGVC datasets. Zero-shot performance is generally unsatisfactory, particularly on specialized domains like Aircraft recognition. When fine-tuning with augmented data generated by our method, we achieve substantial improvements over Diff-Mix, with an average gain of 3.98% across all datasets.

# E    Complete Image Quality Evaluation

In Section 4.3 of the main paper, we presented image quality evaluation on ImageNet and iNaturalist, the two largest and most challenging datasets in our benchmark. For completeness, Table 11 presents the full evaluation across all six remaining FGVC datasets. Our method consistently achieves the best performance across all metrics and datasets. Notably, our approach maintains superior diversity (Vendi Score) and simultaneously improves fidelity (lower DreamSim) and generation accuracy. This demonstrates that the fidelity-diversity trade-off is effectively balanced by our SCA embeddings across different visual domains, from fine-grained flowers and birds to aircraft and food categories.

# F    Comparison with Off-the-Shelf Diffusion Adapters

To contextualize our approach within the broader landscape of image-conditioned diffusion models, we compare our method against several state-of-the-art off-the-shelf adapters: IP-Adapter [89], ELITE [84], E4T [26], and FastComposer [86]. These methods provide pre-trained image encoders and adapter modules that can be directly applied to new domains without task-specific fine-tuning.

| Dataset | Method | DreamSim↓ | Vendi↑ | CLIP-T↑ | Gen. Top-1 | Gen. Top-5 |
|---|---|---|---|---|---|---|
| Flowers | Real-Guidance | 0.3307 | 15.16 | **0.3236** | 87.31% | 95.30% |
| | DA-Fusion | 0.3260 | 13.06 | 0.2795 | 88.24% | 96.22% |
| | Diff-Mix | 0.3195 | 14.20 | N/A | 91.83% | 99.00% |
| | **SCA (Ours)** | **0.3051** | **15.31** | 0.3089 | **95.27%** | **99.88%** |
| Cars | Real-Guidance | 0.4382 | **8.01** | **0.3627** | 54.27% | 82.93% |
| | DA-Fusion | 0.4263 | 7.59 | 0.3169 | 60.85% | 85.60% |
| | Diff-Mix | 0.4035 | 7.72 | N/A | 85.46% | 98.69% |
| | **SCA (Ours)** | **0.3880** | 7.96 | 0.3411 | **86.60%** | **99.07%** |
| Aircraft | Real-Guidance | 0.4676 | 7.59 | **0.3411** | 25.67% | 65.99% |
| | DA-Fusion | 0.4549 | 6.54 | 0.2974 | 33.58% | 71.35% |
| | Diff-Mix | 0.4230 | 7.04 | N/A | 49.46% | 81.96% |
| | **SCA (Ours)** | **0.4001** | **7.61** | 0.3167 | **60.17%** | **92.95%** |
| CUB | Real-Guidance | 0.4584 | **11.91** | **0.3479** | 63.03% | 93.75% |
| | DA-Fusion | 0.4624 | 11.81 | 0.3033 | 62.14% | 90.97% |
| | Diff-Mix | 0.4379 | 11.49 | N/A | 69.53% | 96.49% |
| | **SCA (Ours)** | **0.4092** | 11.89 | 0.3283 | **73.66%** | **97.64%** |
| Food-101 | Real-Guidance | 0.4833 | **21.44** | **0.3413** | 87.71% | 99.03% |
| | DA-Fusion | 0.4796 | 19.46 | 0.2974 | 87.56% | 98.41% |
| | Diff-Mix | 0.4513 | 20.13 | N/A | 90.16% | 99.22% |
| | **SCA (Ours)** | **0.4374** | 21.31 | 0.3289 | **92.48%** | **99.84%** |
| Dogs | Real-Guidance | 0.4731 | 37.92 | **0.3668** | 79.38% | 96.09% |
| | DA-Fusion | 0.4569 | 32.07 | 0.3011 | 79.74% | 97.27% |
| | Diff-Mix | 0.4274 | 34.73 | N/A | 81.24% | 97.40% |
| | **SCA (Ours)** | **0.4105** | **37.99** | 0.3439 | **83.78%** | **98.26%** |

Table 11: Complete image quality evaluation across all FGVC datasets. We report perceptual similarity (DreamSim ↓), sample diversity (Vendi Score ↑), text-image alignment (CLIP-T ↑), and generation accuracy (Gen. Top-1/Top-5). Our SCA method consistently achieves the best balance across all metrics.

| Synthesis Model | Image Encoder | DreamSim↓ | Vendi↑ | CLIP-T↑ | Gen. Top-1 | Gen. Top-5 |
|---|---|---|---|---|---|---|
| ELITE [84] | CLIP-B | 0.5239 | 29.37 | 0.2816 | 20.11% | 30.57% |
| E4T [26] | CLIP-L | 0.5384 | 31.22 | 0.2819 | 2.26% | 5.82% |
| IP-Adapter [89] | CLIP-H | 0.5037 | 31.92 | 0.3007 | 26.18% | 48.28% |
| FastComposer [86] | CLIP-L | 0.5194 | 30.82 | 0.2944 | 4.06% | 9.94% |
| SDXL-Finetuned | CLIP-B | 0.4812 | 31.85 | 0.2721 | 36.53% | 62.00% |
| SDXL-Finetuned | DINOv2-B | 0.4537 | 32.24 | 0.2803 | 58.45% | 80.84% |
| **SDXL-Finetuned (Ours)** | **SCA** | **0.4293** | **34.48** | **0.3106** | **67.28%** | **87.48%** |

Table 12: Comparison with off-the-shelf diffusion adapters on iNaturalist. Our SCA method achieves substantial improvements over pre-trained adapters and alternative visual encoders across all metrics.

Following the evaluation protocol in Table 5 of the main paper, we conduct experiments on the iNaturalist dataset. For each method, we generate synthetic data using the respective adapter and measure: (1) concept preservation using DreamSim [23], (2) sample diversity using the Vendi Score [22], (3) text-image alignment using CLIP-T [61], and (4) generation quality using Top-1 and Top-5 accuracy from an external classifier (a strong model from the TIMM leaderboard). We additionally include ablations using SDXL fine-tuned with CLIP and DINOv2 encoders (without our SCA training procedure) to isolate the contribution of our proposed loss function.

Table 12 presents a comprehensive comparison. Our method consistently outperforms all off-the-shelf adapters across every metric. Notably, we achieve a 41.1% improvement in Gen. Top-1 accuracy over the best off-the-shelf adapter (IP-Adapter: 26.18% vs. Ours: 67.28%), demonstrating the critical importance of domain-specific adaptation for fine-grained visual tasks.

Pre-trained adapters such as IP-Adapter and ELITE, while effective for general-purpose image generation, show poor generation quality on fine-grained classification tasks. Even IP-Adapter with CLIP-H achieves only 26.18% Top-1 accuracy, indicating that the generated images frequently do not preserve the correct fine-grained category. This suggests that generic adapters lack the specialized visual understanding required for subtle inter-class distinctions in domains like species

| Model | IN-LT | iNat | Cars | CUB | Flowers | Aircraft | Food | Dogs | Average |
|---|---|---|---|---|---|---|---|---|---|
| SCA Probing | 69.64 | 65.74 | 94.31 | 90.24 | 99.17 | 85.22 | 93.97 | 90.93 | 86.15 |
| SCA Augmented (Ours) | **79.17** | **74.11** | **95.17** | **91.47** | **99.92** | **88.42** | **95.61** | **91.84** | **89.46** |
| Improvement | +9.53 | +8.37 | +0.86 | +1.23 | +0.75 | +3.20 | +1.64 | +0.91 | +3.31 |

Table 13: Comparison of direct SCA feature classification vs. SCA-guided generative augmentation. Classification accuracy (%) on eight datasets demonstrates that generative augmentation provides substantial improvements over direct feature use.

identification. Fine-tuning SDXL with CLIP-B embeddings (without our SCA training) yields 36.53% Top-1 accuracy, already surpassing all off-the-shelf methods. This demonstrates that task-specific adaptation of the synthesis model is essential for fine-grained domains. Switching to DINOv2-B embeddings further improves performance to 58.45%, highlighting the importance of encoder choice. Our proposed method, which combines DINOv2-based initialization with our angular margin loss training, achieves 67.28% Top-1 accuracy, which is an 8.83% improvement over DINOv2 alone. This improvement, coupled with the best CLIP-T score (0.3106) and highest diversity (34.48 Vendi Score), demonstrates that our SCA training procedure effectively learns to encode salient visual concepts while maintaining text alignment and sample diversity.

# G   Direct Classification with SCA Features

Our approach can be understood through the lens of knowledge distillation: the SCA embedding model encodes task-specific visual priors, which are then transferred to downstream classifiers via the synthesis model. The synthesis model, leveraging its world knowledge, generates images with diverse incidental features (backgrounds, lighting, poses) while preserving salient concepts. This process enables the downstream classifier to learn robust representations that focus on class-discriminative features rather than spurious correlations.

A natural question arises: could we bypass the synthesis stage entirely and use SCA embeddings directly for classification? Despite conceptually appealing, this approach would forfeit the primary benefit of our method that leverages the foundation synthesis model's world knowledge to generate diverse training samples that improve classifier robustness.

To investigate this question empirically, we compare two approaches: (1) **SCA Probing**: training a linear classification head on frozen SCA features, and (2) **SCA Augmented**: our full method using SCA-guided generative augmentation. For fair comparison, we use OpenAI CLIP-B/16 as the backbone for SCA training, matching the initialization used for downstream classifiers in our main experiments. As shown in Table 13, our full generative augmentation method outperforms direct SCA feature classification by an average of 3.31% across eight datasets. The improvement is particularly pronounced on challenging datasets: ImageNet-LT (+9.53%) and iNaturalist (+8.37%), both of which feature long-tailed distributions with limited samples for many classes.

It is important to emphasize that SCA training is not a silver bullet for isolating salient concepts with perfect precision. Rather, it is an effective mitigation strategy that reduces conflicts between image and text prompts during synthesis, helping achieve a better fidelity-diversity trade-off in generated data. The synthesis model remains the primary source of distributional diversity that drives downstream classifier robustness.

