# OpenReview forum: "Salient Concept-Aware Generative Data Augmentation"
_NeurIPS.cc/2025/Conference — NeurIPS 2025 poster_

### Official Review · Reviewer_Vnct · 2025-06-21

**Clarity:** 4
**Significance:** 4
**Originality:** 4
**Rating:** 6
**Confidence:** 3

**Summary:**

This paper addresses the challenge of balancing fidelity and diversity in generative data augmentation (GDA) methods conditioned on both image and text prompts. The key idea is to use a salient concept-aware (SCA) image embedding model that emphasizes class-relevant features while suppressing irrelevant background or context information. The method first independently trains domain-specific salient concept-aware embedding model with a penalty to improve both the intra-class compactness and inter-class discrepancy. Then the method trains the synthesis model, taking frozen image and text embeddings as inputs and only tunes the adapters. This approach does not require intensive subject-specific optimization or additional input processing such as saliency segmentation. Experiments on eight fine-grained visual categorization datasets show improved classification accuracy compared with prior GDA methods, particularly in challenging long-tail and out-of-distribution scenarios. Extensive ablation studies further demonstrate the advantages and stability of the proposed method.

**Questions:**

- In the introduction, the paper refers to synthetic image generation as a form of distillation. Based on this, can generative data augmentation be understood as a process where knowledge from the salient concept-aware embedding model is distilled into the classifier via the synthesis model? From this perspective, I am curious whether using the features from the salient concept-aware embedding model directly for classification would yield results comparable to those obtained through generative data augmentation.

- In the comparisons for conventional, long-tail, and OOD classification, is the amount of synthetic data generated by each method kept consistent? If not, I would appreciate it if the authors could report the amount of synthetic data produced by each method.

**Ethical Concerns:**

["NO or VERY MINOR ethics concerns only"]

**Final Justification:**

I believe all of my concerns have been fully addressed. Therefore, I am willing to raise my score.

**Limitations:**

No. The authors should include technical limitations and potential social impacts.

**Quality:**

4

**Strengths And Weaknesses:**

**Strengths**

- The paper is clearly written and provides valuable high-level insights. The figures in the main text significantly aid understanding, and the supplementary material offers comprehensive implementation details.

- The work identifies a practical and impactful limitation of existing GDA methods—namely, that their embeddings often entangle information that is less relevant to the concept of interest.

- The proposed two-stage training strategy, which first finetunes a domain-specific salient concept-aware embedding model and then trains the synthesis model, is effective. The use of a novel penalty to improve both intra-class compactness and inter-class separation during embedding model finetuning is particularly interesting. The design of the conditional synthesis model and image prompt adapter is also well-motivated, and the paper provides clear strategies for constructing and filtering training data.

- The experiments are thorough, with extensive comparisons across multiple datasets demonstrating state-of-the-art performance. In addition, detailed ablation studies convincingly validate the effectiveness of each component.

**Weaknesses**

- The proposed approach still requires finetuning both the salient concept-aware embedding model and the synthesis model for each new dataset, which limits its generalizability and falls short of achieving truly optimization-free general data augmentation.

---

> ### Author Rebuttal · Authors · 2025-07-31
>
> **Question**: *In the introduction, the paper refers to synthetic image generation as a form of distillation. Based on this, can generative data augmentation be understood as a process where knowledge from the salient concept-aware embedding model is distilled into the classifier via the synthesis model? From this perspective, I am curious whether using the features from the salient concept-aware embedding model directly for classification would yield results comparable to those obtained through generative data augmentation.*
>
> **Answer**: Our work is motivated by practical challenges that researchers often need to train expert classifiers on domain-specific datasets that are typically limited in quantity, class-imbalanced, or require adaptation to new classes with very few examples. To address this problem, we use foundation synthesis model with world knowledge to generate images with diverse incidental features to improve the robustness of the downstream classifier that can focus on the salient concept. We can think of it as a form of indirect knowledge distillation.
>
> However, existing text-to-image or GDA approaches often fail to generate faithful images due to either the limitations of textual descriptions or inability to incorporate task priors. The role of SCA is to introduce the task priors to the synthesis model, achieving optimal fidelity-diversity trade-off. In this way, the foundation synthesis model serves as a task-aware source of world knowledge that the downstream classifier can effectively learn from.
>
> It’s also important to note that the proposed SCA training mechanism is not a silver bullet for learning to isolate salient concepts with exceptional precision. It is an effective mitigation strategy to reduce conflicts between image and text prompts during synthesis, eventually helping achieve a better fidelity-diversity trade-off in the generated data.
>
> Therefore, we suspect that the performance of features from SCA models alone cannot be comparable to that of our augmentation method.
> To validate this, we evaluate the performance of the SCA embedding model by probing its frozen features with classification head. For a fair comparison, we used the OpenAI CLIP‑B/16 backbone for SCA probing (borrowed from the experiment requested by Reviewer MqZv to compare DINO with CLIP), since our downstream classifiers are also trained from OpenAI CLIP‑B/16.
>
> Model               | IN-LT  | iNat   | CAR196 | CUB200 | Flower102 | Aircraft100 | Food101 | Dogs120 | Average
> --------------------|--------|--------|--------|--------|-----------|-------------|----------|--------|----------
> SCA Probing         | 69.64% | 65.74% | 94.31% | 90.24% | 99.17%    | 85.22%      | 93.97%   | 90.93% | 86.15%
> SCA Augmented       | **79.17**% | **74.11**% | **95.17**% | **91.47**% | **99.92**%    | **88.42**%      | **95.61**%   | **91.84**% | **89.46**%
>
> Our augmentation method out-performs the SCA embedding by an average of 3.31% over 8 datasets.
>
> **Question**: *In the comparisons for conventional, long-tail, and OOD classification, is the amount of synthetic data generated by each method kept consistent? If not, I would appreciate it if the authors could report the amount of synthetic data produced by each method.*
>
> **Answer**: For CutMix and Mixup, we apply it to every training sample in the mini-batch.
> For GDA methods, we generated synthetic data with their public code and use them to train our downstream classifiers. The classifier training protocol outlined in section 4.1 is the same for all methods, except for the synthetic-to-real ratio K and the probability of sampling synthetic images P during training. We follow the settings from original papers for the choices of K and P described below:
>
> * For our method, we use P=20% for ImageNet and iNaturalist, and P=50% for all other FGVC dataset. In figure 4 of the manuscript, we show that our method is robust for P in the range from 20% to 50%. We set K=5 all datasets.
> * For Real-Guidance, we use P=50% and K=5 for all datasets.
> * For Da-Fusion, we use P=50% and K=10 for all datasets.
> * For Diff-Mix, we use P=10% and K=5 for all datasets.
>
> In addition, we guarantee that at least 250 synthetic data are generated for each category, applicable to all methods. This is to make sure that enough data are generated for few-shot experiments and under-represented tail categories in LT experiments.
>
>
> **Question**: *The proposed approach still requires finetuning both the salient concept-aware embedding model and the synthesis model for each new dataset, which limits its generalizability and falls short of achieving truly optimization-free general data augmentation.*
>
> **Answer**: We acknowledge that the proposed approach requires fine-tuning both the salient concept-aware embedding model and the synthesis model for each new dataset.
> However, it is important for our method to remain functional for samples from previously unseen or underrepresented classes within the same domain as the training data. For example, if the training dataset contains various dog breeds but excludes bulldogs, the model should still recognize a bulldog as a distinct and relevant class and accurately preserve its concept during the synthesis process. This is partially demonstrated in our LT and OOD experiments in Table 4 of the manuscript, and we will dive deeper in this rebuttal.
>
> We consider an “open-set” setup where our embedding and synthesis models were trained using only 80% of randomly selected species from the iNat training set, and the remaining 20% of species were held out and only used for evaluation. We compare our method with two peer personalized image generation methods as recommended by Reviewer MqZv.
>
> Dataset        | DreamSim↓ | Vendi↑ | CLIPT↑  | Gen.Top1  | Gen.Top5
> ---------------|-----------|--------|---------|-----------|----------
> ELITE [2]      | 0.5226    | 29.15  | 0.2837  | 14.10%    | 25.47%
> IP-Adapter [1] | 0.4993    | 31.80  | 0.2993  | 23.76%    | 48.34%
> Ours           | **0.4419**    | **34.28**  | **0.3033**  | **60.35**%    | **84.19**%
>
> The Top-5 accuracy of the external timm leaderboard classifier on the generated images (Gen. Top5) drops moderately from 87.48% (in Table 5 of the manuscript) to 84.19%, primarily due to the challenge of generating samples for unseen classes. However, despite this constraint, our approach still outperforms all peer methods across concept preservation and text-image alignment metrics, demonstrating its superior ability to extrapolate fine-grained visual concepts beyond the training distribution. This aligns with findings from our long-tail and OOD experiments in Section 4.2, where our model demonstrated strong performance for underrepresented/unseen categories.
>
> *[1] IP-Adapter: Text Compatible Image Prompt Adapter for Text-to-Image Diffusion Models, Ye et al.*
>
> *[2] ELITE: Encoding Visual Concepts into Textual Embeddings for Customized Text-to-Image Generation, Wei et al.*

---

> > ### Comment · Reviewer_Vnct · 2025-08-03
> >
> > Thank you to the authors for their response and the detailed experimental results. I believe all of my concerns have been fully addressed. Therefore, I am willing to raise my score.

---

### Official Review · Reviewer_MqZv · 2025-07-01

**Clarity:** 3
**Significance:** 2
**Originality:** 2
**Rating:** 4
**Confidence:** 2

**Summary:**

The paper presents an improved method for generative data augmentation that focuses on generating diverse images of the same class while preserving only the relevant class features, while discarding unimportant information, such as the background or position.
The method is based on fine-tuning a diffusion model that is conditioned on "salient concept-aware" image embeddings, allowing the model to generate diverse images of the same input class.
The method is validated through different evaluations, including conventional and few-shot classification.

**Questions:**

Some things are not completely clear:
- Is the Domain Expert Embedding Model trained together with the diffusion model?
- Is it trained separately for each task?
- L. 128-131: Why CLIP Image features are not suitable for the task? Unlike DINO, CLIP is trained under contrastive settings between images and **text** such that it captures the important concepts in the image.
- I wonder how the method improves over augmentation that utilizes other pre-trained diffusion adapters, e.g.:
[1] IP-Adapter: Text Compatible Image Prompt Adapter for Text-to-Image Diffusion Models, Ye et al.
[2] BLIP-Diffusion: Pre-trained Subject Representation for Controllable Text-to-Image Generation and Editing, Li et al.
[3] ELITE: Encoding Visual Concepts into Textual Embeddings for Customized Text-to-Image Generation, Wei et al.

**Ethical Concerns:**

["NO or VERY MINOR ethics concerns only"]

**Final Justification:**

My concerns have been addressed in the rebuttal and I raise my score to 4.

**Limitations:**

Mentioned limitations were not discussed.

**Paper Formatting Concerns:**

I haven't noticed major formatting issues.

**Quality:**

3

**Strengths And Weaknesses:**

The method is simple and effective in classification tasks.
However, the method is expensive to train; IIUIC, it requires fine-tuning of the SCA and the diffusion model for each classification task.
Comparison to other off-the-shelf adapters is required.
Additionally, I found the scope of the method to be limited: could the proposed data augmentation help to improve generative personalization tasks (e.g., 1, 2), or other discriminative tasks?

[1] An Image is Worth One Word: Personalizing Text-to-Image Generation using Textual Inversion, Gal et al.
[2] Multi-Concept Customization of Text-to-Image Diffusion, Kumari et al.

---

> ### Author Rebuttal · Authors · 2025-07-31
>
> **Question**: *However, the method is expensive to train; IIUIC, it requires fine-tuning of the SCA and the diffusion model for each classification task.*
>
> **Answer**: We acknowledge that our approach is more computationally intensive than classical augmentation or existing generative data augmentation (GDA) methods, as it involves training a SCA embedding model and fine-tuning a diffusion model. As shown in the table below, on an 8×A100 GPU setup, training takes approximately 20–30 minutes for smaller fine-grained datasets (e.g., ~10K images), and up to 3 days for large-scale datasets such as iNaturalist 2021 (2.7M images).
> We argue that this cost is practical for most application use-cases, especially for scenarios where curating high-quality in-domain data is prohibitively expensive. Our method offers a promising solution, giving consistent performance improvements over baseline GDA methods.
>
> Dataset | FGVC       | iNat2018 | ImageNet1K | iNat2021
> --------|------------|----------|------------|------------
> Size    | ~10K       | 440K     | 1.3M       | 2.7M
> Time    | 20-30 mins | 19 hours | 25 hours   | ~3 days
>
> **Question**: *Comparison to other off-the-shelf adapters is required.*
>
> **Question**: *I wonder how the method improves over augmentation that utilizes other pre-trained diffusion adapters, e.g.: [1] IP-Adapter [2] BLIP-Diffusion [3] ELITE*
>
> **Answer**: We compare our method with IP-Adapter [1], ELITE [3], E4T [4], , FastComposer [5].
>
> Dataset | Synthesis Model  | Embedder  | DreamSim↓ | Vendi↑  | CLIPT↑  | Gen. Top1 | Gen. Top5
> --------|------------------|-----------|-----------|---------|---------|-----------|------------
> iNat    | ELITE [3]           | Clip-B    | 0.5239    | 29.37   | 0.2816  | 20.11%    | 30.57%
> iNat    | E4T [4]             | Clip-L    | 0.5384    | 31.22   | 0.2819  | 2.26%     | 5.82%
> iNat    | IP-Adapter [1]       | Clip-H    | 0.5037    | 31.92   | 0.3007  | 26.18%    | 48.28%
> iNat    | FastComposer [5]     | Clip-L    | 0.5194    | 30.82   | 0.2944  | 4.06%     | 9.94%
> iNat    | SDXL-Finetuned   | Clip-B    | 0.4812    | 31.85   | 0.2721  | 36.53%    | 62.00%
> iNat    | SDXL-Finetuned   | DINOv2-B  | 0.4537    | 32.24   | 0.2803  | 58.45%    | 80.84%
> iNat    | SDXL-Finetuned   | SCA       | **0.4293**    | **34.48**   | **0.3106**  | **67.28**%    | **87.48**%
>
> In line with Table 5 of the manuscript, we conduct evaluation using the iNaturalist dataset. For each method, we generated synthetic data and measured concept preservation using DreamSim, diversity with Vendi, and text–image alignment using CLIP-T. Additionally, we evaluate Top-1 and Top-5 accuracy on the synthetic data using an external classifier from the TIMM leaderboard (denoted as Gen. Top-1 and Gen. Top-5). Our method consistently outperforms other off-the-shelf adapters across all metrics, with 41.1% and 2.56 improvements on Gen. Top-1 accuracy and CLIPT scores, respectively.
>
> *[1] Ye et al., “IP-Adapter: Text-Compatible Image Prompt Adapter for Text-to-Image Diffusion Models,” arXiv:2308.06721, 2023.*
>
> *[3] Wei et al., “ELITE: Encoding Visual Concepts into Textual Embeddings for Customized Text-to-Image Generation,” ICCV 2023.*
>
> *[4] Gal et al., “Encoder-Based Domain Tuning for Fast Personalization of Text-to-Image Models,” TOG 2023.*
>
> *[5] Xiao et al., “FastComposer: Tuning-Free Multi-Subject Image Generation with Localized Attention,” arXiv:2305.10431, 2023.*
>
> **Question**: *Is the Domain Expert Embedding Model trained together with the diffusion model?*
>
> **Answer**: The domain expert embedding model is trained separately and kept frozen during the diffusion model fine-tuning.
>
> **Question**: *Is it trained separately for each task?*
>
> **Answer**: Yes, the SCA embedding model as well as the diffusion model are trained separately for each task.
>
> **Question**: *L. 128-131: Why CLIP Image features are not suitable for the task? Unlike DINO, CLIP is trained under contrastive settings between images and text such that it captures the important concepts in the image.*
>
> **Answer**: Our experimental findings show that DINOv2 consistently outperforms CLIP by a significant margin across all eight datasets. We include the iNaturalist results alongside the off-the-shelf adapter results in the table above. Here we fine-tune the synthesis model to be compatible with each image embedding model, following the same procedure used with SCA. By switching from DINOv2 to Clip, the CLIPT score is dropped by 0.0082 and the Gen. Top-1 score is dropped by 21.92%. We suspect the reason is that CLIP is optimized for global semantic consistency with high-level language descriptions, making it less effective for fine-grained visual tasks such as image generation that require understanding of low-level details. In contrast, DINOv2 is trained with a self-supervised objective targeted for vision task only. This observation is also supported by recent work [6], where they show in Table 2 that aligning diffusion model representations with DINO embeddings leads to superior performance in generative tasks, compared to CLIP embeddings.
>
> *[6] Zhang et al., “Representation Alignment for Generation: Training Diffusion Transformers is Easier Than You Think,” ICLR 2025.*
>
> **Question**: *Additionally, I found the scope of the method to be limited: could the proposed data augmentation help to improve generative personalization tasks (e.g., 1, 2), or other discriminative tasks?*
>
> **Answer**:
> [Generative Personalization Tasks] Our method could potentially improve personalized image generation methods that rely on image encoders to extract image prompt information (e.g., [1], [3–5]). Prior work has mainly focused on redesigning multimodal attention in diffusion models, and our approach is orthogonal to them as we improve the image encoder to reduce conflicts between irrelevant visual features and text prompts, resulting in images that more faithfully align with the intended concepts.
>
> Due to the new rebuttal policy, we cannot include qualitative comparisons. Informally, we observe that our method achieves better text‑prompt alignment than IP‑Adapter and greater visual quality than ELITE. For example, given the prompt "Brilliant purple flower, aged brick wall, gentle shadows" and an input image of a violet on green grass, ELITE outputs a pink-purple blend which does not look like flowers. IP-Adapter outputs a counterfactual image of a violet on a stone road with surrounding green grass. Our method faithfully generates a violet near a red brick wall under sunlight.
>
> The primary focus of this paper is on discriminative tasks, motivated by practical challenges in niche domains where expert models are only trained on small or class‑imbalanced domain-specific datasets, a regime where our method shines. Extending our approach to generative personalization is an exciting direction for future work.
>
> [Discriminative Vision Tasks] Our current synthesis pipeline is built on a diffusion model (SDXL), which is not well-suited for complex vision tasks such as text-rich VQA or GraphQA. There are other interesting experiments for our method to demonstrate: a) broader applicability to diverse domains, b) application to multi-modal models.
>
> **a) broader applicability to diverse domains**:
> We perform a few-shot experiment on Places365, a dataset for abstract scene understanding rather than the object-centric datasets we primarily focused on. The evaluation is done on its validation set. We follow the same the N-way 10-shot experiment setting in Table 2 of the manuscript. The result is shown below:
>
> Method    | Vanilla | CutMix  | Mixup  | Da-Fusion | Diff-Mix  | Ours
> ----------|---------|---------|--------|-----------|-----------|-------
> Top-1     | 35.32   | 36.23   | 35.53  | 36.57     | 37.53     | **39.06**
> Top-5     | 64.15   | 65.62   | 64.47  | 65.88     | 66.91     | **68.69**
>
> We report classification accuracy (%) using 10-shot training samples on Places 365 validation set. Our method out-performs peer method by 1.53% and 1.78% in Top-1 and 5 accuracy, respectively, demonstrating that our method is also effective on datasets of abstract scene as well.
>
> **b) application to multi-modal models**:
> Classification is still an important task for modern multi-modal models that often require fine-tuning domain-specific dataset to achieve production-level performance. To demonstrate the effectiveness of our generation method in this context, we explore multi-modal visual classification by performing supervised fine-tuning on an MLLM using FGVC datasets. In this setup, we tune only the projector and vision encoder and keep the LLM frozen. The classification problem is reformulated as a 26-choice QA task, where the ground-truth label is always included among the choices. An prompt example is provided below:
> *You are an expert classifier. Based on the visual evidence in the image, select the most appropriate category. Please choose ONE of the following:
> A. beef_tartare
> B. creme_brulee
> ...
> Z. tacos
> Please respond ONLY with your answer wrapped in this format: \<answer\> LETTER \<\/answer\>.*
>
> Method                | CUB     | Aircrafts | Flowers  | Cars    | Dogs    | Food   | Average
> ----------------------|---------|-----------|----------|---------|---------|--------|--------
> Zero-shot             | 79.44%  | 58.78%    | 82.91%   | 87.58%  | 78.48%  | 90.31% | 79.58%
> Fine-tuned - Diff-Mix | 89.37%  | 66.92%    | 94.71%   | 95.30%  | 87.14%  | 83.23% | 86.11%
> Fine-tuned - Ours     | **90.49**%  | **79.31**%    | **95.30**%   | **96.57**%  | **87.33**%  | **91.53**% | **90.09**%
>
> We report classification accuracy (%) under zero-shot and N-way 10-shot setting with Qwen2.5-VL-7B backbone. The zero-shot performance is not satisfactory on majority of the FGVC datasets. When fine-tuning the model with augmented data, our method out-performs Diff-mix by 3.98% on average across 6 FGVC datasets.

---

### Official Review · Reviewer_fpDS · 2025-07-02

**Clarity:** 3
**Significance:** 3
**Originality:** 2
**Rating:** 5
**Confidence:** 4

**Summary:**

This paper proposes an image generation framework for data augmentation on image and text prompts. The main component of the design is a "salient concept-aware" image embedding model. The idea is to preserve key attributes of the input image while reducing the influence of the background features. This is achieved by training the image embeddings with additional penalty terms. Experiments are conducted on eight fine-grained image datasets, and the proposed method is compared with other data augmentation methods.

**Questions:**

- Figure 2: for image (c), it seems the proposed model SCA learns some sort of style transfer--for the given image (f), it treats the black mask as the texture of the fly and draws a black fly/moth in image (c). This is slightly different from what is expected if the learned embeddings are truly about either the fly or the background flowers. Can the authors clarify what's happening here when a salient object is masked out?

- Please see the discussion on Weaknesses and clarify how the proposed loss term enables the learning of 'salient' features.

- In some experiments (such as Section 4.3), only a small subset of the dataset is selected. Is there a reason not to test on all fine-grained datasets?

Overall, I like the simplicity of the approach and, in contrast, how much improvement it can achieve in the long-tail classification tasks. It shows promising results that might inspire more future research in addressing data augmentation for long-tail classes.

**Ethical Concerns:**

["NO or VERY MINOR ethics concerns only"]

**Final Justification:**

After reading the rebuttal, I think the paper is in a better shape with the addition of the experiments and clarifications. I recommend the acceptance of the paper.

**Limitations:**

No discussion on limitations.

**Quality:**

3

**Strengths And Weaknesses:**

### Strengths
- Generative data augmentation is an important task. Trying to address it to generate fine-grained categories is a good motivation.
- The experiment section is fairly comprehensive, covering a wide range of methods and benchmarks, as well as essential ablations.
- The paper focuses on obtaining better image embeddings, without the need to make substantial changes to the generation model itself. This makes the approach flexible, and it is possible to adapt to other generation models.

### Weaknesses
- The core of this paper is the loss term in the Salient Concept-Aware Embedding Model (Eq. 1). It looks like a tuned version of a contrastive loss (based on cosine similarity with margins) in a contrastive learning framework. It is unclear to me where the 'salient' part comes from in this formulation: I think it is closer to a max-margin positive feature vs negative feature learning scheme.
- The 0.73% accuracy improvement in the conventional setting is underwhelming. One would expect that having a good data augmentation to obtain more diverse data should improve the results more, even in the traditional setting without the focus on the few-shot or long-tail cases.

---

> ### Author Rebuttal · Authors · 2025-07-31
>
> **Question**: *The core of this paper is the loss term in the Salient Concept-Aware Embedding Model (Eq. 1). It looks like a tuned version of a contrastive loss (based on cosine similarity with margins) in a contrastive learning framework. It is unclear to me where the 'salient' part comes from in this formulation: I think it is closer to a max-margin positive feature vs negative feature learning scheme.*
>
> **Answer**: Salient concept learning is from the combination of the proposed loss and the structure of the training data. For example, consider two images of the same bird species, one standing on a tree, the other on a rock. Cross-entropy loss is happy as long as the prediction is correct, even if the model encodes incidental features (e.g., trees or rocks). In contrast, our margin-based loss encourages intra-class compactness in the embedding space, penalizing incidental variations that introduce noise.
>
> We would like to note that SCA is not a silver bullet for isolating salient concepts. It is an effective mitigation strategy to reduce conflicts between image and text prompts during synthesis, eventually helping achieve a better fidelity-diversity trade-off in the generated data.
>
> We agree with the reviewer that our formulation can be viewed as a form of max-margin learning. But we want to highlight the role of the angular loss: image embeddings are first normalized before being fed into the adapter and cross-attention layers of the diffusion model. This normalization stabilizes gradient flow during fine-tuning and better aligns the image embeddings with the text embeddings. Our angular margin–based loss is specifically designed to optimize these normalized embeddings (see Lines 144–145 of the manuscript). In contrast, embeddings trained with cross-entropy typically require post-hoc normalization and gives inferior performance in our experiments (see Table 6 of the manuscript).
>
> **Question**: *The 0.73% accuracy improvement in the conventional setting is underwhelming. One would expect that having a good data augmentation to obtain more diverse data should improve the results more, even in the traditional setting without the focus on the few-shot or long-tail cases.*
>
> **Answer**: We agree with the reviewer that a 0.73% improvement over peer methods in the conventional setting is modest in absolute terms. Another perspective is to compare relative gains: the current SOTA GDA method improves performance over the vanilla baseline by 0.855%, and our method achieves a 1.585% improvement. This is an 85% relative performance gain over the SOTA method, which is not marginal.
>
> It is also true that the conventional setting is more challenging to improve: with fixed task complexity, as datasets become more balanced and diversified, there is less room for robustness improvement through augmentation (e.g., a model pretrained on web-scale data may show little to no further gain on the easy task distinguishing cats and dogs). Our work is motivated by practical challenges in niche domains, where expert models must be trained on domain-specific datasets that are often small, class-imbalanced, or updated incrementally (e.g., with new categories over time). In these settings, our method is a good choice for practitioners as it consistently outperforms baseline GDA methods across conventional, few-shot, long-tail, and out-of-distribution classification tasks.
>
> **Question**: *Figure 2: for image (c), it seems the proposed model SCA learns some sort of style transfer—for the given image (f), it treats the black mask as the texture of the fly and draws a black fly/moth in image (c). This is slightly different from what is expected if the learned embeddings are truly about either the fly or the background flowers. Can the authors clarify what's happening here when a salient object is masked out?*
>
> **Answer**:
> This is a very interesting observation that we didn’t pay attention to before. We suspect that the SCA model is influenced by learned priors from the training data, driving it to interpret the black masked region at the center into one of the species from the iNaturalist dataset. To validate this we computed the SCA embedding of the masked image (Fig. 2(f)) and compared it with the average class embeddings across all species in iNat dataset. The top nearest matches were:
>
> 1. Atolmis rubricollis (Insects), cosine similarity: 0.31
> 2. Lucidota atra (Insects), cosine similarity: 0.19
>
> Both of them are black-colored insect species. Given this result, we agree with the reviewer that our model’s behavior is analogous to style transfer. Furthermore, we think the SCA embedding encodes key foreground information from the masked image (Fig. 2(f)) including style, color, textual of the concept of a dark insect. Conditioned on those, the synthesis model is able to reconstruct the final generated image (Fig. 2(c)).
>
>
> **Question**: *In some experiments (such as Section 4.3), only a small subset of the dataset is selected. Is there a reason not to test on all fine-grained datasets?*
>
> **Answer**: We used the two largest, representative and most challenging datasets (ImageNet and iNaturalist) in our experiment setting to analyze the quality of the generated images in Table 5 of Section 4.3.
> We complete the Table 5 with the remaining 6 FGVC datasets. We will include the following results in the supplementary of the revised manuscript.
>
>
> Dataset   | Model         | DreamSim↓ | Vendi↑  | CLIPT↑  | Gen. Top1 | Gen. Top5
> ----------|---------------|-----------|---------|---------|-----------|------------
> Flowers   | Real-Guidance | 0.3307    | 15.16   | **0.3236**  | 87.31%    | 95.30%
> Flowers  | DA-Fusion     | 0.3260    | 13.06   | 0.2795  | 88.24%    | 96.22%
> Flowers  | Diff-Mix      | 0.3195    | 14.20   | N/A     | 91.83     | 99.00%
> Flowers  | SCA           | **0.3051**    | **15.31**   | 0.3089  | **95.27**%    | **99.88**%
> &nbsp;    |               |           |         |         |           |
> Cars      | Real-Guidance | 0.4382    | **8.01**    | **0.3627**  | 54.27%    | 82.93%
> Cars      | DA-Fusion     | 0.4263    | 7.59    | 0.3169  | 60.85%    | 85.60%
> Cars      | Diff-Mix      | 0.4035    | 7.72    | N/A     | 85.46%    | 98.69%
> Cars      | SCA           | **0.3880**    | 7.96    | 0.3411  | **86.60**%    | **99.07**%
> &nbsp;    |               |           |         |         |           |
> Aircrafts | Real-Guidance | 0.4676    | 7.59    | **0.3411**  | 25.67%    | 65.99%
> Aircrafts | DA-Fusion     | 0.4549    | 6.54    | 0.2974  | 33.58%    | 71.35%
> Aircrafts | Diff-Mix      | 0.4230    | 7.04    | N/A     | 49.46%    | 81.96%
> Aircrafts | SCA           | **0.4001**    | **7.61**    | 0.3167  | **60.17**%    | **92.95**%
> &nbsp;    |               |           |         |         |           |
> CUB       | Real-Guidance | 0.4584    | **11.91**   | **0.3479**  | 63.03%    | 93.75%
> CUB       | DA-Fusion     | 0.4624    | 11.81   | 0.3033  | 62.14%    | 90.97%
> CUB       | Diff-Mix.     | 0.4379    | 11.49   | N/A     | 69.53%    | 96.49%
> CUB       | SCA           | **0.4092**    | 11.89   | 0.3283  | **73.66**%    | **97.64**%
> &nbsp;    |               |           |         |         |           |
> Food      | Real-Guidance | 0.4833    | **21.44**   | **0.3413**  | 87.71%    | 99.03%
> Food      | DA-Fusion     | 0.4796    | 19.46   | 0.2974  | 87.56%    | 98.41%
> Food      | Diff-Mix.     | 0.4513    | 20.13   | N/A     | 90.16%    | 99.22%
> Food      | SCA           | **0.4374**    | 21.31   | 0.3289  | **92.48**%    | **99.84**%
> &nbsp;    |               |           |         |         |           |
> Dogs      | Real-Guidance | 0.4731    | 37.92   | **0.3668**  | 79.38%    | 96.09%
> Dogs      | DA-Fusion     | 0.4569    | 32.07   | 0.3011  | 79.74%    | 97.27%
> Dogs      | Diff-Mix      | 0.4274    | 34.73   | N/A     | 81.24%    | 97.40%
> Dogs      | SCA           | **0.4105**    | **37.99**   | 0.3439  | **83.78**%    | **98.26**%

---

> > ### Comment · Reviewer_fpDS · 2025-08-06
> >
> > Thanks for the response. The addition of experiments on more datasets is important, please add them to the final version. Overall, I think this paper proposes a valuable approach, so I recommend its acceptance.

---

### Official Review · Reviewer_wJKD · 2025-07-04

**Clarity:** 4
**Significance:** 4
**Originality:** 3
**Rating:** 4
**Confidence:** 2

**Summary:**

This paper presents a framework for generative data augmentation (GDA) that effectively addresses the fidelity-diversity trade-off by introducing a salient concept-aware (SCA) image embedding model. The proposed method disentangles essential visual features from irrelevant attributes, enabling better alignment with text prompts while preserving class-discriminative details. The authors demonstrate significant improvements in classification accuracy across multiple fine-grained datasets, particularly in long-tail settings.

**Questions:**

According to the W.1-2.

**Ethical Concerns:**

["NO or VERY MINOR ethics concerns only"]

**Quality:**

3

**Strengths And Weaknesses:**

Strengths:

1. Proposes a novel salient concept-aware (SCA) embedding model that effectively disentangles essential class-discriminative features from irrelevant image attributes (e.g., background), directly addressing the fidelity-diversity trade-off in generative data augmentation.
2. Eliminates reliance on manual interventions (e.g., segmentation masks) or subject-specific optimization, enhancing scalability for real-world applications.
3. Demonstrates state-of-the-art results across eight fine-grained datasets under diverse settings (conventional, long-tail, few-shot), with significant accuracy gains (up to +6.5%).

Weaknesses:
This paper does not seem to have any major flaws.
1. Focuses exclusively on fine-grained visual tasks; applicability to broader or non-visual domains remains unverified.
2. Fails to analyze computational costs of training the SCA embedding model, a critical factor for adoption.

---

> ### Author Rebuttal · Authors · 2025-07-31
>
> **Question**: *Focuses exclusively on fine-grained visual tasks; applicability to broader or non-visual domains remains unverified.*
>
> **Answer**: Our focus on fine-grained classification tasks is motivated by practical challenges in niche domains where expert models must be trained on domain-specific datasets that are often small or class-imbalanced. GDA is particularly valuable in these settings, as it can generate faithful and diverse samples by leveraging world priors learned from synthesis models, thereby improving robustness of downstream classifier. However, our current synthesis pipeline is built on a diffusion model (SDXL), which is not well-suited for complex vision tasks such as text-rich VQA or GraphQA. There are other interesting experiments for our method to demonstrate: a) broader applicability to diverse domains, b) application to multi-modal models.
>
> **a) broader applicability to diverse domains**:
> We perform a few-shot experiment on Places365, a dataset for abstract scene understanding rather than the object-centric datasets we primarily focused on. The evaluation is done on its validation set. We follow the same the N-way 10-shot experiment setting in Table 2 of the manuscript. The result is shown below:
>
> Method    | Vanilla | CutMix  | Mixup  | Da-Fusion | Diff-Mix  | Ours
> ----------|---------|---------|--------|-----------|-----------|-------
> Top-1     | 35.32   | 36.23   | 35.53  | 36.57     | 37.53     | **39.06**
> Top-5     | 64.15   | 65.62   | 64.47  | 65.88     | 66.91     | **68.69**
>
> *Table 1: Classification accuracy (%) using 10-shot training samples on Places 365 validation set. Our method out-performs peer method by 1.53% and 1.78% in Top-1 and 5 accuracy, respectively, demonstrating that our method is also effective on datasets of abstract scene as well.*
>
> **b) application to multi-modal models**:
> Classification is still an important task for modern multi-modal models that often require fine-tuning domain-specific dataset to achieve production-level performance. To demonstrate the effectiveness of our generation method in this context, we explore multi-modal visual classification by performing supervised fine-tuning on an MLLM using FGVC datasets. In this setup, we tune only the projector and vision encoder and keep the LLM frozen. The classification problem is reformulated as a 26-choice QA task, where the ground-truth label is always included among the choices. An prompt example is provided below:
> *"You are an expert classifier. Based on the visual evidence in the image, select the most appropriate category. Please choose ONE of the following:
> A. beef_tartare
> B. creme_brulee
> ...
> Z. tacos
> Please respond ONLY with your answer wrapped in this format: \<answer\> LETTER \<\/answer\>."*
>
> Method                | CUB     | Aircrafts | Flowers  | Cars    | Dogs    | Food   | Average
> ----------------------|---------|-----------|----------|---------|---------|--------|--------
> Zero-shot             | 79.44%  | 58.78%    | 82.91%   | 87.58%  | 78.48%  | 90.31% | 79.58%
> Fine-tuned - Diff-Mix | 89.37%  | 66.92%    | 94.71%   | 95.30%  | 87.14%  | 83.23% | 86.11%
> Fine-tuned - Ours     | **90.49**%  | **79.31**%    | **95.30**%   | **96.57**%  | **87.33**%  | **91.53**% | **90.09**%
>
> *Table 2: Classification accuracy (%) under N-way 10-shot setting with Qwen2.5-VL-7B backbone. We report the zero-shot and fine-tuned performance. The zero-shot performance is not satisfactory on majority of the FGVC datasets. When fine-tuning the model with augmented data, our method out-performs Diff-mix by 3.98% on average across 6 FGVC datasets.*
>
>
> **Question**: *Fails to analyze computational costs of training the SCA embedding model, a critical factor for adoption.*
>
> **Answer**: Our method involves two training stages: (1) training the SCA embedding model, and (2) fine-tuning the synthesis model using SCA embeddings; training the SCA model requires about 20% of the time needed to fine-tune the synthesis model. As shown in the table below, on an 8×A100 GPU setup, training both components takes approximately 20–30 minutes for smaller-scale FGVC datasets of size ~10K, and up to 3 days for iNat2021 with 2.7M images, the largest dataset we have experimented with.  We argue that this cost is practical for most application use-cases, especially for scenarios where curating high-quality in-domain data is prohibitively expensive. Our method offers a promising solution, giving consistent performance improvements over baseline GDA methods.
>
> Dataset | FGVC       | iNat2018 | ImageNet1K | iNat2021
> --------|------------|----------|------------|------------
> Size    | ~10K       | 440K     | 1.3M       | 2.7M
> Time    | 20-30 mins | 19 hours | 25 hours   | ~3 days

---

> > ### Comment · Reviewer_wJKD · 2025-08-08
> > **Official Comment by Reviewer wJKD**
> >
> > The authors have clearly and adequately addressed my concerns, and the efforts to provide the new results are appreciated. I recommend this paper for acceptance.

---

### Note · Authors · 2025-08-11

We thank all reviewers for the feedback. We are encouraged by the strong consensus toward acceptance from **Reviewer wJKD**, **Reviewer fpDS**, and **Reviewer Vnct**. We have also carefully addressed all concerns from **Reviewer MqZv**, including computational practicality, comparisons with other adapters, and applicability to other tasks.

Given that no reviewer has unresolved objections or further questions, we will summarize the main additions and clarifications made during rebuttal:

**Broader applicability**

In response to **Reviewer wJKD**, we conducted experiments on new tasks, including abstract scene classification (Places365) and multi-modal visual classification, achieving consistent gains over baselines. In response to **Reviewer MqZv**, we compared with personalized image generation methods (IP-Adapter, ELITE, E4T, FastComposer), where our method consistently outperformed them in concept preservation, diversity, and text–image alignment.

**Additional datasets**

As requested by **Reviewer fpDS**, we completed results for the remaining FGVC datasets for Table 5 of Section 4.3, which will be included in the supplementary material of the revised manuscript.

**Clarifications**

In response to **Reviewer Vnct**, we clarified the role of the SCA loss in learning salient concepts and its contribution to the overall pipeline, supported by quantitative evidence.

**Open-set capability**

In response to **Reviewer Vnct**, we extended experiments to evaluate performance on unseen but related categories, showing that our synthesis model can generate faithful images in these settings, further justifying its effectiveness in long-tail and OOD scenarios.

**Remark**

Our work is motivated by practical challenges in niche domains, where expert models must be trained on domain-specific datasets that are often small or class-imbalanced. We highlight the importance of the vision encoder in mitigating the fidelity–diversity trade-off in generative data augmentation, and provide an effective, practical approach for improving model performance when collecting additional high-quality data is challenging or cost-prohibitive. We hope the clarifications and results convey the significance of our contribution for publication. Thank you very much for your attention.

---

### Decision · Program_Chairs · 2025-09-17

**Decision:**

Accept (poster)

**Comment:**

The paper proposes a generative data augmentation method that balances fidelity and diversity when generating synthetic training images from image and text prompts. The approach first trains a domain-specific embedding model to extract salient class features from individual images, then fine-tunes a diffusion model conditioned on these embeddings. This allows for generating diverse images while preserving input concept fidelity.

Reviewers noted strengths, including strong empirical results, especially in long-tail settings, a comprehensive evaluation across eight datasets, and clear presentation. Main weaknesses include computational costs from dataset-specific training and modest gains in standard settings. The rebuttal provided additional experiments on FGVC and scene data, a computational cost analysis, and more baseline comparisons. All reviewers found the responses satisfactory and recommend acceptance.

The Area Chair agrees with the reviewers and recommends acceptance.